

# motoRneuron: an open-source R toolbox for time-domain motor unit analyses

Andrew J. Tweedell[1] and Matthew S. Tenan[2]

[1] Human Research and Engineering Directorate, United States Army Research Laboratory, Aberdeen Proving Ground, MD, United States of America
[2] Defense Health Agency, Falls Church, VA, United States of America

Corresponding author
Andrew J. Tweedell,
andrew.j.tweedell.civ@mail.mil

## ABSTRACT

Motor unit synchronization is the tendency of motor neurons and their associated muscle fibers to discharge near-simultaneously. It has been theorized as a control mechanism for force generation by common excitatory inputs to these motor neurons. Magnitude of synchronization is calculated from peaks in cross-correlation histograms between motor unit discharge trains. However, there are many different methods for detecting these peaks and even more indices for calculating synchronization from them. Methodology is diverse, typically laboratory-specific and requires expensive software, like Matlab or LabView. This lack of standardization makes it difficult to draw definitive conclusions about motor unit synchronization. A free, open-source toolbox, "motoRneuron", for the R programming language, has been developed which contains functions for calculating time domain synchronization using different methods found in the literature. The objective of this paper is to detail the toolbox's functionality and present a case study showing how the same synchronization index can differ when different methods are used to compute it. A pair of motor unit action potential trains were collected from the forearm during a isometric finger flexion task using fine wire electromyography. The motoRneuron package was used to analyze the discharge time of the motor units for time-domain synchronization. The primary function "mu_synch" automatically performed the cross-correlation analysis using three different peak detection methods, the cumulative sum method, the $z$-score method, and a subjective visual method. As function parameters defined by the user, only first order recurrence intervals were calculated and a 1 ms bin width was used to create the cross correlation histogram. Output from the function were six common synchronization indices, the common input strength (CIS), $k'$, $k'-1$, E, S, and Synch Index. In general, there was a high degree of synchronization between the two motor units. However, there was a varying degree of synchronization between methods. For example, the widely used CIS index, which represents a rate of synchronized discharges, shows a 45% difference between the visual and $z$-score methods. This singular example demonstrates how a lack of consensus in motor unit synchronization methodologies may lead to substantially differing results between studies. The motoRneuron toolbox provides researchers with a standard interface and software to examine time-domain motor unit synchronization.

## INTRODUCTION

Motor unit synchronization is the tendency of separate motor units (i.e., motor neurons and their associated muscle fibers) to discharge near-simultaneously (within 1–5 ms of each other) more often than would be expected by chance (*Farmer et al., 1997*; *Semmler, 2002*). Similar to a frequency-domain coherence analysis, these time-domain analyses are often interpreted as indicators of functional connectivity between motor neurons through common excitatory post-synaptic potentials (*Sears & Stagg, 1976*). Typically, cross-correlation analyses are employed, whereby the discharge times of one motor unit are correlated against those of another concurrently active motor unit (Fig. 1) and a histogram is created based on these recurrence intervals. Peaks in the histogram represent a higher probability of a discharge from the response motor unit around that latency of the reference motor unit discharge (seen in Fig. 2B). Various indices are calculated from these peaks and their magnitude indicates the level of synchronization (for review see *Farmer et al., 1997*; *Semmler, 2002*; *Farina & Negro, 2015*), which appear to be a critical factor in force modulation. For example, synchronous activation of muscle fibers produce longer and greater twitch forces than if they were activated asynchronously (*Merton, 1954*). In practice, this phenomenon is evidenced in strength-trained individuals, who display higher motor unit synchrony than untrained individuals do (*Semmler & Nordstrom, 1998*; *Fling, Christie & Kamen, 2009*). Although beneficial for producing high forces, synchronization has been shown to be detrimental to force steadiness (*Yao, Fuglevand & Enoka, 2000*). Thus, understanding motor unit synchronization seems to be important for modeling neuromuscular performance.

Over the last few decades, it has become much easier and cheaper to collect motor unit action potentials with either intramuscular or decomposed surface electromyography. Researchers have gone from examining synchronization in 2–3 motor units to 15+ in a single contraction (*Schmied & Descarreaux, 2010*; *Defreitas et al., 2014*). Unfortunately, while data collection technology has improved and multiplied, so have the options for synchronization analysis. Reconciling results from different types of analyses remains difficult. Concerning the cross-correlation analysis, there are numerous ways in which to determine the size and location of peaks present in histograms. Methodology is largely laboratory specific, with some groups using automated methods like the $z$-score method or the cumulative sum method. Before automated methods were developed, subjective, visual analysis was used. Within these methods, parameters such as the number of orders of recurrence intervals used and histogram bin size are likely to vary as well. Additionally, there are a number of indices available to characterize synchronization magnitude. Common input strength (CIS) and $k'$ ("$k$ prime") are most often reported; however, the Synch Index (SI) and others are available. The lack of standardization with respect to motor unit synchronization hinders our ability to make definitive conclusions. Therefore, we have developed the open-source toolbox "motoRneuron" in the statistical programming language R (henceforth referred to as R) for the calculation of time-domain synchronization using various peak determination methods. This toolbox provides a list of functions to
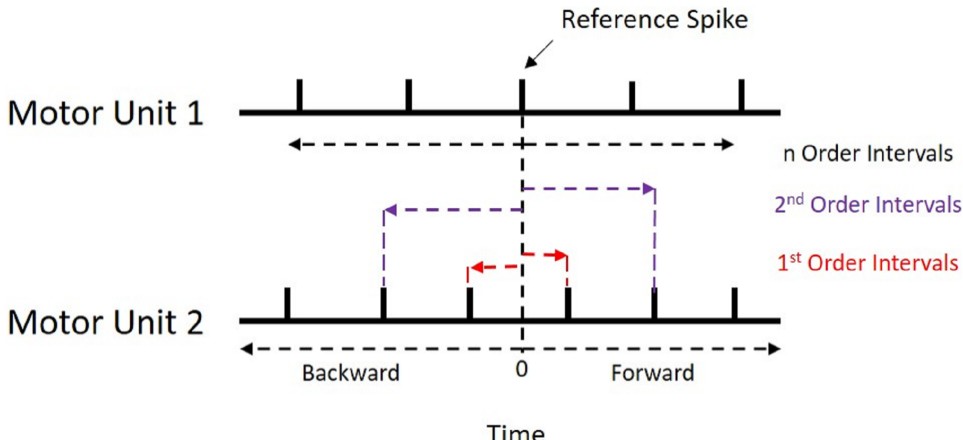

**Figure 1** **Recurrence Intervals Diagram.** Schematic representation of the recurrence intervals between two concurrently active motor units. Each discharge from one motor unit is used as a reference point to determine forward and backward latencies to the discharges of the second motor unit. The first order intervals are the latencies to the first forward and backward discharges (noted in red). The second order are the second forward and backward discharges (noted in purple).

calculate recurrence intervals, create and plot cross-correlation histograms, and ultimately, calculate synchronization indices.

R has quickly risen in popularity recently because of its very active user/developer base that rapidly iterates to improve the functionality of the language. Typically, programs that perform synchronization analyses are handed down through laboratories using paid software like Matlab and LabView. Meanwhile, R and our toolbox are freely available. Source code for all R functions are available to the user. MotoRneuron was created as a free, open-source platform with which users can perform all necessary functions to calculate synchronization, or alter to suit their unique needs. With the numerous ways to calculate synchronization, it is unlikely all methods will yield the same results. This toolbox allows for better standardization of techniques and for more comprehensive data mining in the motor control community. The primary objective of this paper is to detail the functionality and demonstrate the use of the motoRneuron toolbox for investigating motor unit time-domain synchronization. The secondary objective is to present a case study showing how much the same synchronization index can differ when different methods are used to compute it.

## METHODS

### Participant

The example motor unit discharge data included in the R package and subsequently analyzed for this paper was selected from a single participant during a previous study investigating finger flexion during a simulated trigger pull task. For the study, participants had no known neurological or metabolic disease and provided informed consent in accordance with the United States Army Research Laboratory Institutional

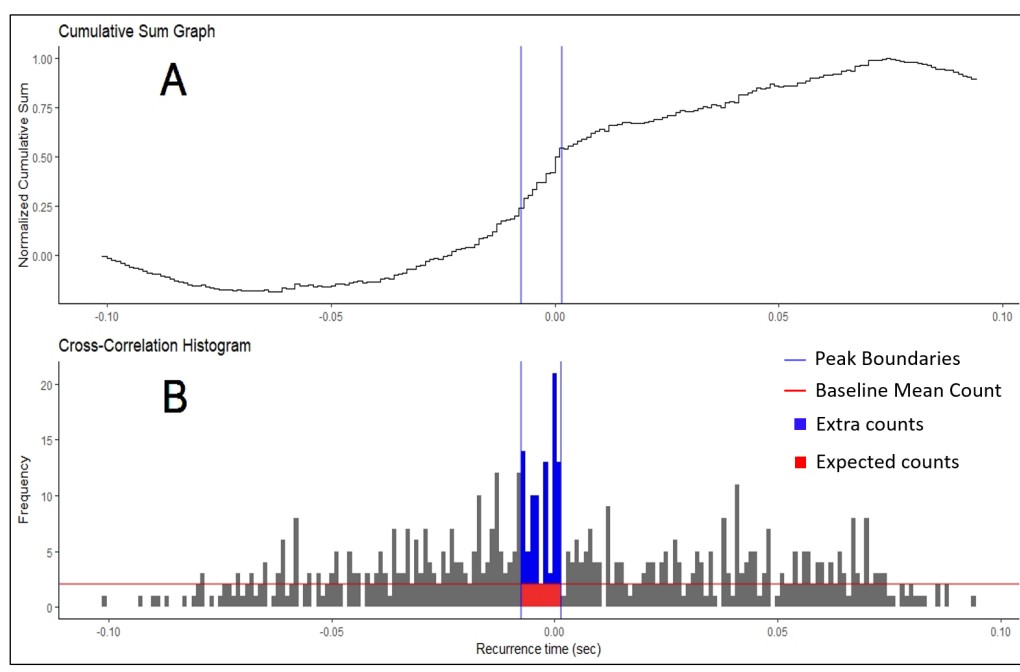

**Figure 2  Cumulative Sum and Histogram.** (A) Example of cumulative sum graph of bin counts. (B) Cross-correlation histogram rendered in RStudio using the "ggplot" package. Peak boundaries (blue lines) were determined by visual analysis of the cumulative sum graph, where a peak is seen as a deflection near time 0. Synchronization indices are calculated based on the relationship between the counts of the histogram expected due to chance (in red) and the counts that are in excess of what is expected (in blue). In most cases, this threshold for determination (red line) is the baseline mean count of the histogram.

Review Board (approval number ARL 16-099). Data collection methods are briefly described here.

## Motor unit data collection

Motor unit discharge data was collected using fine wire electromyography (EMG) during an isometric finger flexion task. The participant was fitted into an apparatus with a handgrip, which maintains wrist supination, and a load cell embedded in it to obtain flexion forces generated by the first phalange (pointer finger). After familiarization, three maximal voluntary isometric contractions (MVIC) were performed to determine the 20% MVIC used for the isometric task. For more detail on the experimental set up, please see the work by *Haynes et al. (2018)*. Fine-wire electrodes (Natus® Neurology, Wisconsin, USA) were then inserted with the use of an ultrasound guide into the flexor digitorum superficialis (FDS) muscle. Motor unit synchronization has been shown to be higher in smaller, distal muscles compared to larger, proximal movement muscles (*Keen et al., 2012*). A pre-gelled sticker electrode (B&L Engineering, Santa Ana, California, USA) was placed on the first phalange metacarpal phalangeal joint as a ground. Force data was sampled at 1 kHz while the EMG was sampled at 20 kHz. A ramp-and-hold isometric contraction paradigm was performed at 20% MVIC for 30 s.

## Data reduction

Discharge data was only used for the steady force portion of the contraction and processed offline, using Spike2 (Cambridge Electronic Design Limited, Cambridge, England, UK). The raw EMG signal was decomposed into motor unit action potential trains using a template matching technique (*Cambridge Electronic Design Limited, 2018*). Briefly, a potential motor unit action potential was manually chosen from the beginning of the contraction. The waveform was extracted and a new template was created with a width of ±35% of the chosen waveform's amplitude. A threshold was then set such that whenever the signal crossed it (i.e., a potential motor unit action potential); the waveform was compared against the current template. Spikes match a template if more than 60% of the points in a spike fall within the template. The motor unit action potential spike trains were subsequently manually checked for accuracy. Time series of discharge times of two separate motor unit action potential trains were then down sampled to 1 kHz.

## Synchronization analysis

All synchronization analyses were performed with the custom R package "motoRneuron" in RStudio (v 1.1.453) using R (v 3.5.0) (*R Core Team, 2015*). The general steps to determine time-domain synchronization are described in the introduction; however, more detail is provided here. First, the discharge times of the paired motor units were cross-correlated. The motor unit with fewer discharges was called the reference unit, while the other was referred to as the event unit. The time differences between each action potential of the reference unit and the nearest forward and nearest backward action potential of the event unit were calculated and are referred to as recurrence intervals (see Fig. 1 for an illustrative example). The first order intervals refer to the latencies of the first forward and backward discharges (noted in red in Fig. 1). The second order intervals are the second forward and backward discharges (noted in purple in Fig. 1), and so on. Only first order intervals were used for the current investigation. Multi-order recurrence intervals can be calculated; however, some researchers argue that only first order intervals should be used for analysis, as the presence of harmonics within the cross-correlation may cause non-physiological peaks to appear in the long latency portions of the histogram (*De Luca, Roy & Erim, 1993*). These intervals were discretized into 1-ms bins for a cross-correlation histogram (Fig. 2B).

## Peak determination

The three methods employed in this toolbox reflect the three broad classes of cross-correlation histogram peak determination in the current motor unit synchronization literature. These methods comes from the notion that if motor unit action potential trains are independent, then there is no relationship between the firing of one motor unit and the firing of the other and each latency or recurrence interval (e.g., −2 ms, +1 ms, or +6 ms) has an equal probability of occurring within plus or minus the mean inter-spike interval for the slower motor unit (*De Luca, Roy & Erim, 1993*). If the intervals are sufficiently random, a resulting cross correlation histogram represents

a discrete uniform probability distribution where the probability of an interval falling within a bin is constant across the histogram. It should appear flat with small variations in individual bin count. The intent of the methods is to find the bins in the experimental histograms that are significantly greater than what is expected by this random chance.

The oldest method for finding these boundary bins is through visual determination (*Datta & Stephens, 1990*; *Nordstrom, Fuglevand & Enoka, 1992*; *Schmied & Descarreaux, 2010*). Only bins from the −100 to +100 ms region of the histogram were used in analysis. First, the baseline bin count was calculated as the mean bin count of the region less than −60 and more than 60 ms (i.e., bins from −100 to −60 ms and 60 to 100 ms). This baseline bin count was subsequently subtracted from each original bin count. Then, the new bin counts were progressively summed across the histogram. Each bin in this cumulative sum was then divided by the baseline mean count of the original histogram to produce a "normalized" cumulative sum graph (*Ellaway, 1978*). Large increases or decreases in bin counts are seen as deflections in the graph. Large, positive inflects near time 0 may indicate greater synchronization and are judged by an investigator. An example of this plot is shown in Fig. 2A, with boundaries chosen by a user highlighted.

The next method, demonstrated by *Keen & Fuglevand (2004)* and *Keen et al. (2012)*, also uses the same initial cumulative sum (cumsum) technique across the bins from −100 to +100 ms. However, peak boundaries were determined algorithmically. First, the maximum and minimum bin count values of the cumulative sum graph were found and subtracted to calculate the range in values. Peak boundaries were considered the bins in the histogram that corresponded to 10 and 90% of this range. As a caveat, this method assumes that the baseline sections of the histogram (i.e., the non-peak regions on the outside) remain relatively stable or flat. If there is considerable variability in bin count of the baseline region, the 10 and 90% bins may not fall within the expected deflection near time 0 and may cause an unreasonably large peak width. The peak was considered significant if its mean bin count exceeds the sum of the mean and 1.96 times the standard deviation of the baseline bins (again considered as −100 to −60 ms and 60 to 100 ms). If no significant peak was detected, synchronization indices were calculated from the ±5 ms region of the histogram. When analyzing multiple pairs for synchronization, it is recommended that users use caution when interpreting peak location and width as this ±5 ms default may obscure the actual peak when aggregated together.

The third peak determination method included is the $z$-score technique (*Contessa, Adam & De Luca, 2009*; *Defreitas et al., 2014*). In this method, a 95% confidence interval is calculated from a "shuffled" version of the experimental histogram. This shuffled histogram represents the correlation of the two motor unit trains if they were independent. The parameters taken from the experimental histogram to create this new histogram are the range (taken as ± mean interspike interval (ISI)) and the number of recurrence intervals (or total counts in the histogram). Then, a random sample equal in size to the number of recurrence intervals from the experimental histogram is taken from between ± mean ISI. This sample is drawn from a uniform distribution where each value between ± mean ISI

has an equal probability of being picked. The resulting sample is then binned just like the original data to create this "shuffled" histogram. Because it is randomly sampled, there is still variability in the bin counts. The technique essentially redistributes the recurrence intervals evenly across the entire histogram. The justification for reshuffling is that it keeps the probabilistic nature of the data (i.e., variation in bin count) but removes the interdependence of the point processes so as to test the null hypothesis that the distribution is uniform (*De Luca, Roy & Erim, 1993*). Because of this assumption, we consider each bin count a Bernoulli event with a binomial distribution and each bin can be assessed against the same confidence interval. The mean and standard deviation of bin counts of this shuffled histogram is used to calculate this 95% confidence interval (Eq. (1)).

$$\text{Significance Threshold} = \bar{x} + \left( 1.96 \times \sqrt{\frac{\sum (x - \bar{x})^2}{n}} \right). \tag{1}$$

Any bins in the experimental histogram within $\pm 10$ ms of 0 above this threshold were considered to be significantly greater than expected due to chance and subsequently used for analysis. If no peak was detected, synchronization indices were returned as 0. Because the $z$-score method tests each bin between $\pm 10$ ms individually, peak bins are not necessarily adjacent. In addition, the peak location and duration will be constant.

## Synchronization indices

Once the boundaries for each peak determination method were established, six commonly used synchronization indices were calculated (*Nordstrom, Fuglevand & Enoka, 1992*; *De Luca, Roy & Erim, 1993*; *Kamen & Roy, 2000*). The peak of the histogram consisted of two different regions; the region of counts that are expected due to chance and the region containing "extra" counts more than what is expected due to chance (Fig. 2B). These extra counts are the number of counts in the peak bins over a certain threshold. In the $Z$-score method, this is the significance threshold calculated from the shuffled histogram. In the Visual method, the threshold is the baseline mean bin count. The total counts in peak consists of the summation of the regions. Synchronization indices quantify the relationship between these different regions. The CIS index is commonly used because it allows for normalization with respect to trial duration. *Nordstrom, Fuglevand & Enoka (1992)* developed the CIS because most other indices available at that time were influenced by discharge rate. The $k'$ index represents the ratio of all counts in the peak to the expected counts, often calculated as the baseline mean. The $k' - 1$ is similar but only includes the counts considered extra, or more than expected due to chance. The E and S indices represent the ratio of the extra counts in the peak and the number of discharges from the reference unit and both units, respectively. The Synch Index (SI) is very similar to the E index; however, it is the ratio of the extra counts in the peak to half the total counts in the histogram. Their specific equations are listed below. The larger the magnitude of the indices, the higher the chances that the motor units are firing in synchronization.

$$CIS = \frac{\text{extra counts in peak}}{\text{duration of trial (s)}}$$

$$k' = \frac{\text{total counts in peak}}{\text{expected counts in peak}}$$

$$k' - 1 = \frac{\text{extra counts in peak}}{\text{expected counts in peak}}$$

$$E = \frac{\text{extra counts in peak}}{\text{number of discharges from reference motor unit}}$$

$$S = \frac{\text{extra counts in peak}}{\text{total number of discharges from both motor units}}$$

$$SI = \frac{\text{extra counts in peak}}{\frac{\text{total counts}}{2}}.$$

Along with the synchronization indices listed above, the peak significance, duration and peak center were also calculated. These refer to the $z$-score of the peak, the width of the peak and the bin location of the center of the peak, respectively, to help characterize the latency of synchronization.

**MotoRneuron package implementation**

This section details the implementation of the motoRneuron package that was used for synchronization analysis. MotoRneuron was configured on a Windows 10 computer (Enterprise V. 1703. Intel® Core $^{TM}$ 2.80 GHz, x64-based processor). The toolbox is under the GNU General Public License version 2. Example code is provided throughout to instruct readers on its use. In addition, help files for specific details about their functions are included within the package itself. It is highly recommended to download R and RStudio in order to follow along with the sample scripts provided. Briefly, R uses command-line scripting to perform functions on data within the working environment. Common data formats for R are vectors, matrices, lists, and data frames, which can be imported into the working environment from any number of formats, including text or csv files. MotoRneuron leverages many functions not included in base R which are automatically incorporated by downloading the following add-on packages from Github or the Comprehensive R Archive Network (CRAN): 'dplyr', 'ggplot2', 'dygraphs', 'magrittr', and 'tseries' (*Milton-Bache & Wickham, 2014*; *Trapletti & Hornik, 2018*; *Vanderkam et al., 2018*; *Wickham et al., 2018a*; *Wickham et al., 2018b*).

To access motoRneuron through R and all the functions, sample data, and help files wherein, the following functions are called in the console of RStudio. "*install.packages*" will automatically download the package from CRAN. "*library*" will attach the packages items to your working environment for use.

```
> install.packages(''motoRneuron'')
> library(motoRneuron)
```

The code repository can also be found in Github at the following URL: https://github.com/tweedell/motoRneuron.

Motor unit data collected for this manuscript is automatically included in the package and accessible with the following code.

```
> Sample_data <- motoRneuron::motor_unit_data
```

The data format is a data frame time series of two concurrently active motor units, named motor_unit_1 and motor_unit_2. Here we provide the code to read in the data and reduce into the constituent motor units discharge times for further use in the package.

```
> motor_unit_1 <- subset(Sample_data, select = Time, motor_unit_1 ==
1)
> motor_unit_1 <- as.vector(motor_unit_1$Time)
> motor_unit_2 <- as.vector(subset(Sample_data, select = Time,
motor_unit_2 ==1)
> motor_unit_2 <- as.vector(motor_unit_2$Time)
```

Below is the output from R for the two motor unit discharge vectors showing the first six time points by calling the function *head*. For example, the first three discharge times for *motor_unit_1* are at 0.035, 0.115, and 0.183 s, while *motor_unit_2* discharged at 0.1, 0.205, and 0.298 s.

```
> head(motor_unit_1)
## [1] 0.035 0.115 0.183 0.250 0.306 0.377 ...
> head(motor_unit_2)
## [1] 0.100 0.205 0.298 0.377 0.471 0.577 ...
```

The primary function of motoRneuron to analyze these motor units is *mu_synch*, which completes all the steps previously described above. The function's syntax and six formal arguments are:

*mu_synch(motor_unit_1, motor_unit_2, method, order, binwidth, plot)*

*Motor_unit_1* and *motor_unit_2* arguments are the vectors of the discharge times of two motor unit action potential trains. The distinction between the reference and event unit is made automatically within the function and is output as a part of the motor unit characteristics. *Method* indicates which method(s) of cross-correlation peak determination is to be used, while *order* and *binwidth* specifies how many orders of recurrence intervals to calculate and the size of the bins for the histogram, respectively. The default argument of *order* is set at 1, indicating only first order intervals are to be used; however, the function is flexible enough to handle any order input by the user. Additionally, the *binwidth* argument is set at a default of 0.001 s or 1 ms. This allows for appropriate resolution in short-term synchronization measurements. The following R scripts was used to call the *mu_synch* function to perform all three methods for first order recurrence intervals with a bin size of 1 ms.

```
> Results <- mu_synch(motor_unit_1, motor_unit_2, method = c(''Visual'',
''Zscore'', ''Cumsum''), order = 1, binwidth = 0.001, plot = FALSE)
```

Each individual method can also be called separately with their own respective functions. *Visual_mu_synch, Cumsum_mu_synch,* and *Zscore_mu_synch* use the same formal arguments as *mu_synch*, except for *method*. What is returned from these functions is a list of individual motor unit characteristic data along with a list of all synchronization indices. Characteristics for each motor unit included are the number of discharges, the

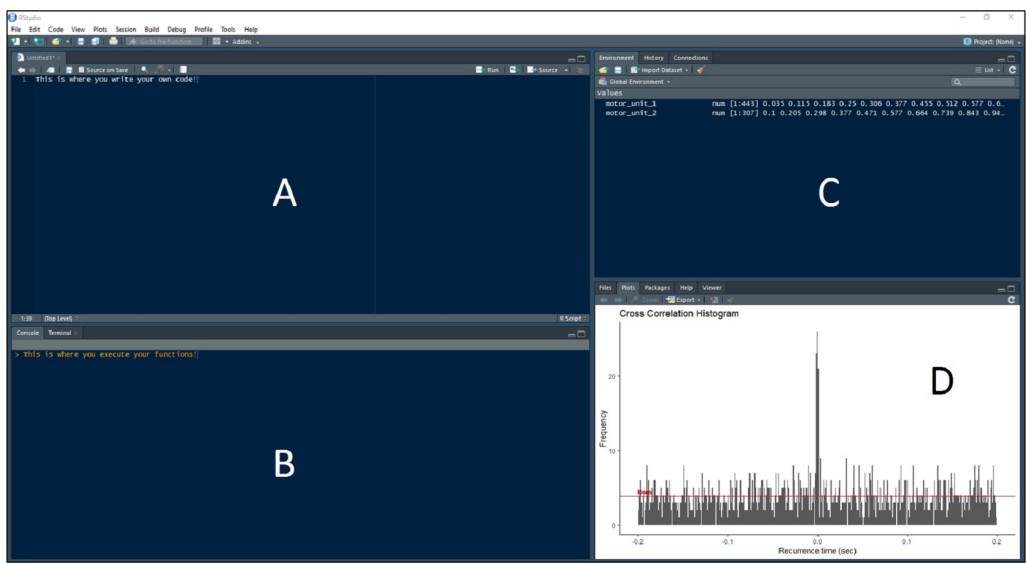

**Figure 3  RStudio Graphical User Interface.** R integrated development environment RStudio's graphical user interface. The interface is made up of four panels: (A) R script panel, (B) Console, (C) Global Environment, (D) plot panel depicting an example cross-correlation histogram.

mean interspike interval (ISI), all ISI's, and the intervals for each specified recurrence order. The plot argument takes a TRUE or FALSE to indicate whether the resulting histogram will be displayed or not.

*Recurrence_intervals* and *bin* are support functions used within the synchronization functions to compute the recurrence intervals and discretize the data for the histogram, but they can also be called separately for individual use. A *plot_bins* function is also available that will display the associated histogram in the Plot window of RStudio (Fig. 3D). This is useful for visually checking data for abnormalities prior to calculating synchronization. The code below creates an R list named '*first_order_intervals*' that contains the motor unit characteristic data along with the first order recurrence intervals.

```
> first_order_intervals <- recurrence_intervals(motor_unit_1,
motor_unit_2, order = 1)
```

To access just the intervals, use the '$' operator to index them. Below, the *head* function is used again just to view the first six elements of the first order intervals.

```
> head(first_order_intervals$'1')
## [1] −0.065 0.015 −0.022 0.045 −0.048 0.008 ...
```

Now these intervals are input to the *bin* function, along with the user-defined bin width, to discretize the intervals into bins for the detection peaks. A data frame '*binned_data*' is created with the code using a bin width of 1 ms. The resulting data frame contains a column depicting the bin, or the amount of time in second before (negative) or after (positive) the reference motor unit discharge, and the frequency of occurrence at that interval. This data frame can be put directly into the *plot_bins* function to display the histogram (such as in Fig. 3D).

Table 1 **Motor unit synchronization indices from the flexor digitorumsuperficialis.** Time-domain synchronization indices calculated from a pair of motor unit action potential trains collected from the flexor digitorum superficialis during a 20% isometric finger flexion task.

| Synchronization index | Visual | Cumsum | Z-score |
|---|---|---|---|
| CIS (impulses/s) | 2.70 | 2.16 | 1.71 |
| $k'$ | 4.64 | 3.80 | 4.29 |
| $k' - 1$ | 3.64 | 2.80 | 3.29 |
| E | 0.26 | 0.21 | 0.17 |
| S | 0.11 | 0.09 | 0.07 |
| SI (%) | 0.26 | 0.21 | 0.17 |
| Peak Duration (s) | 0.010 | 0.010 | 0.011 |
| Peak Center (s) | −0.004 | 0 | 0 |

```
> binned_data <- bin(first_order_intervals$'1', binwidth = 0.001)
> head(binned_data)
## Bin Freq
1 -0.101 1
2 -0.100 0
3 -0.099 0
4 -0.098 0
5 -0.097 0
6 -0.096 0
> plot_bins(binned_data)
```

## RESULTS

The pair of flexor digitorum superficialis motor units tested for synchronization were concurrently active for 29.9 of the 30-second isometric, trigger-finger flexion task. According to the results of the *mu_synch* function, the first motor unit discharged 443 times, with a mean ISI of 0.068 s. The second motor unit discharged 307 times, with a mean ISI of 0.098 s. Synchronization indices and peak characteristics are displayed in Table 1.

## DISCUSSION

The example motor unit pair in the current study analyzed by the new motoRneuron R package demonstrated a high level of synchronization overall. For comparison, *Keen et al. (2012)* used the cumsum method for motor unit pairs in the FDS as well, and found the mean CIS and $k'$ indices (mean ± sd) to be 0.4 ± 0.21 impulse per second and 1.55 ± 0.34, respectively. While differences in individual results from previous studies like this are useful, these comparisons are difficult with such disparate methodologies and indices. Indeed, results from the current study suggest different methods may produce different interpretations about synchronization. For example, for the commonly cited CIS, the Visual method indicated an increase of approximately 1 impulse per second compared to the Z-score method. This 1 impulse per second difference represents 45% difference between the visual and z-score methods of CIS in the FDS. This singular example

demonstrates how a lack of consensus in MU synchronization methodologies may lead to substantially differing results between studies.

There are, however, limitations to using motoRneuron and to interpreting motor unit synchronization in general. The primary limitation is that the package is currently only developed to analyze single pairs of motor units. Some researchers argue that the notion that common input strength can be inferred from time-domain synchronization is overreaching and that synchronization is just a product of firing rate characteristics (*Kline & De Luca, 2016*). Still other researchers postulate that single motor unit analyses are too narrow of a view, as a muscle can consist of hundreds of motor units. They argue a population-based approach should be taken to look at the common drive within the motor neuron pool as the primary determinate of force output (*Farina & Negro, 2015*). In the future, the package can be expanded to address such questions about motor neuron pool common drive with coherence analyses or frequency-domain characterization. The next limitation is that the number of motor unit firing instances, and subsequently the number of recurrence intervals, greatly affects the shape of the cross-correlation histogram. If there are too few, an accurate picture of peak significant cannot be made. This can also been seen in the current experimental data (Fig. 2). The number of recurrence intervals used to generate histograms in previous studies has been reported in the range of approximately 300–3,000 (*Schmied & Descarreaux, 2010*; *Defreitas et al., 2014*; *Schmied, Forget & Vedel, 2014*). While no specific limit has been provided within the literature, it is recommended that visual inspections of the histograms be made to ensure adequate data quality. Finally, while the program and packages discussed in the paper are freely available, synchronization analyses still requires the use of expensive hardware and software to collect and decompose electromyography. As these technologies become cheaper, sources like R will likely play a bigger role in facilitating better analyses.

MotoRneuron is written in the R programming language, which provides an open-source platform to perform data and statistical analysis on motor unit data. The package also allows for the visualization of these analyses through R's powerful and flexible graphics capabilities. An advantage to R, as alluded to before, is its robust statistical computing. Using the R environment allows for direct access to many statistical packages. The "stats" package comes included in base R so many statistical tests are immediately available for testing synchronization metrics. This eliminates the need for transforming and importing data into 3rd party statistical software, such as SAS and SPSS. Simple tests such as $t$-tests and ANOVAS are common, while more complex, multi-level models are available. Bugs or errors in software are common in open-source software like R. As this is the first stable version of the motoRneuron package, it is possible that users will notice performance issues or errors stemming from R version fragmentation or other sources. Users are urged to email any errors or issues found in motoRneuron to the package maintainer (Andrew Tweedell andrew.j.tweedell.civ@mail.mil). Errors that can be fixed will be updated in new versions of the package as they are found. As such, it is important to update the package continually to guarantee efficient performance.

## CONCLUSION

MotoRneuron is a free R package containing a list of functions capable of performing many different cross-correlation analyses for calculating time-domain synchronization metrics for use in the motor control field. The detailed steps from this paper enables researchers to easily examine the many options for calculating and reporting synchronization indices. The example real-world motor unit data provided suggests these different methods contribute to differences in measures of synchronization. With this package, new data can be quickly reconciled with results from previous studies for better physiologic interpretation.

### Citation

Researchers using motoRneuron in a published paper should cite this article and indicate the used version of the package.

## ACKNOWLEDGEMENTS

The authors would like to acknowledge Dr. Courtney Haynes for her work as technical reviewer for the manuscript. The authors would also like to acknowledge the entire R community for providing a free platform for the creation and distribution of this package to the greater scientific community.

### Funding

This work was funded by the Department of Defense Human Systems Integration (Cybernetics) research line at the U.S. Army Research Laboratory (74-A-HRCYB). The funders had no role in study design, data collection and analysis, decision to publish, or preparation of the manuscript.

### Grant Disclosures

The following grant information was disclosed by the authors:
Department of Defense Human Systems Integration (Cybernetics) research line at the U.S. Army Research Laboratory: 74-A-HRCY.

### Competing Interests

Andrew Tweedell is an employee of the United States Department of Defense. Matthew Tenan is an employee of Defense Health Agency, a 3rd party contractor for the United States Department of Defense.

### Author Contributions

- Andrew J. Tweedell conceived and designed the experiments, performed the experiments, analyzed the data, contributed reagents/materials/analysis tools, prepared figures and/or tables, authored or reviewed drafts of the paper, approved the final draft.
- Matthew S. Tenan conceived and designed the experiments, performed the experiments, authored or reviewed drafts of the paper, approved the final draft, reviewed code/programming for computational and syntax errors.

## Human Ethics

The following information was supplied relating to ethical approvals (i.e., approving body and any reference numbers):

The United States Army Research Laboratory—Institutional Review Board approved this study (ARL 16-099).

## Data Availability

Data is available at GitHub: https://github.com/tweedell/motoRneuron or CRAN: https://cran.r-project.org/web/packages/motoRneuron/index.html.

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
