# Peer review of "motoRneuron: an open-source R toolbox for time-domain motor unit analyses"

_PeerJ, doi:10.7717/peerj.7907_

## Round 0.1 · original submission · Major Revisions

While an open-source R toolbox for time-domain motor unit analyses is of clear benefit to the research community, the criteria for publication in PeerJ are that the functionality of the software should, where appropriate, be validated using real-world biological data and/or compared to existing tools. I would therefore like to ask to the authors to include a comparison to existing methods in a revised manuscript. If the authors are happy to revise their manuscript accordingly, I will invite the reviewers to assess the revised manuscript against this criteria.

·

Basic reporting

The article is generally well written and easy to follow. Appropriate literature references are provided, and the context and scope of the article are discussed. All requirements regarding the provided subject data are fulfilled. My main concern regarding the overall structure is that the current interweaving of methodological description, toolbox description, and example R code makes it difficult to find methodological details.

1. While I appreciate the authors' considerable effort put into making the toolbox accessible to new R users, I would prefer a clear distinction between the presentation and discussion of the implemented methods, and example code. Currently, these two aspects are strongly overlapping in the text body, which can make it hard to find the technical details of a method. Would it be an option to separate these matters - mathematical description of the implemented methods on the one hand, and description of their implementation and example code on the other hand - into distinct sections of the article?

2. Generally, references are given where necessary. Still, I would have appreciated more references in several sections. In particular:
a) In the discussion, you write that frequency domain methods will be added in the future. If these are relevant, they should also be mentioned in the introduction.
b) For the different peak determination methods, there is one reference given for each of them. However, it is currently not obvious what exactly this reference refers to - e.g., in line 188, the reference (Keen et al. 2012) could refer either to the whole paragraph or only to the method of using a default peak. If you implemented the whole method following this one reference, please state that clearly in the beginning of the paragraph.
c) Also, as these peak determination methods appear to be known in the literature, it would be useful to provide some references to works (other than the original paper) where these methods have been used.
d) While I am not an expert in this field, it appears to me that there might be newer references that should be cited, e.g., Kline and De Luca (2016): "Synchronization of motor unit firings: an epiphenomenon of firing rate characteristics not common inputs."

3. I am not sure whether Figure 3 is of any utility to the reader and would advocate to remove it.

4. Is equation numbering with letters (line 196) journal standard? It seems very unusual and looks odd (also, because the equation number is to the left of and close to the equation).

5. There seems to be some mistake in the code listing in lines 277-292. Or are all the "$"s and "`"s intentionally there?

6. Several minor errors / suggestions:
Lines 70-71: Something is wrong or missing in this sentence.
Line 85: "The objective..." of this article/paper?
Line 154: "Recurrences" should be "recurrence" (singular).
Line 302: R is not a "scripted code" but a software.
Line 320: "flexibility" should be "flexible".

Experimental design

The authors describe a toolbox which implements several standard methods currently used in the field of motor unit synchronization analysis. The toolbox, including a sample data set, is freely available and open source, and the authors' laudable aim is to further reproducibility and standardization in this field. My main concern regards the presentation of the technical methods implemented in the toolbox, which is currently very brief and would benefit from more detailed description.

There are some details of the implemented methods that are not clear to me. While I understand that this is not a methodological paper, I believe that a precise explanation of the methods implemented in the toolbox is crucial, preferably using standard mathematical notation instead of textual descriptions (e.g. in line 196). In particular:
a) How exactly is the cumsum plot generated? How is it possible that the cumulative sum of a purely positive variable (bin counts divided by mean count) decreases at any point? (See Fig. 2A. and also the example in the toolbox.)
b) What exactly does "(considered as <=-60 and >=60ms)" mean? (Lines 176-177, 187)
c) Determining the peak boundaries as the 10%-90% range (line 185) seems counterintuitive - looking at the graph in Fig. 2A, I would expect this to yield a very unreasonable peak width. Could you elaborate and/or add an example application of this method?
d) What is the justification for assuming a default peak (line 188) if no significant peak can be detected? Doesn't this distort the subsequent analyses?
e) I do not understand the current description of the z-score method. What kind of uniform distribution do you mean? Uniform over which range? How is this related to a "shuffled" histogram? What does it all have to do with z scores? (Lines 191-193)
f) Is it really true that the histogram of independently firing motor units is expected to be completely "flat" (line 193)? I would assume the distribution to be falling off for large first-order distances, as it is e.g. very unlikely that the closest discharge time of the second MU is >1s away.
g) Related to f): Is there a justification for the choice of the +/-100ms range (e.g. line 175)? Is it reasonable across all muscles, subjects and conditions? Is it possible to adjust this value in the toolbox?
h) Is your usage of the term "cross-correlation" actually correct? At no point in the analysis, a correlation seems to be calculated, and the plots are also not what is typically understood as a "correlation diagram".
i) The definition of the "order" of an interval can currently only be found in the caption of figure 1. It should also be part of the main text.

Validity of the findings

I commend the authors on making their toolbox very easily available and accessible, including the provided sample data set. I encountered two issues while following the examples provided in the paper:

1) The command "motor_unit_1 <- as.vector(subset(Sample_data, select = Time, motor_unit_1 == 1))" returned something of type list (not vector), and the subsequent call to "mu_synch" failed with an error message ("'motor_unit_1' and 'motor_unit_2' must be vectors."). I did "motor_unit_1 = motor_unit_1$Time" and the same for MU 2, and that fixed the issue.
2) The call "mu_synch(motor_unit_1, motor_unit_2, method = c("Zscore", "Cumsum"), order = 1, binwidth = 0.001, plot = TRUE)" did not result in any plot being shown.

Additional comments

I commend you for making these efforts to increase reproducibility and standardization of studies in this field. The toolbox looks useful and very well-documented, and you made it very easy to get started using it. If you can fix the current technical limitations of the paper, this will be a useful contribution.

·

Basic reporting

no comment

Experimental design

Some experimental data is presented (Fig. 2 and 3), but the origin of this data is not explained. Although it is not the primary scope of the data, I would suggest a brief explanation of how the data presented was obtained. Also, a more systematic presentation of the values obtained for the different synchronization indexes in the presented data should be included.
A good example may be the McGill et al. "EMGLAB: an interactive EMG decomposition program" J Neurosci Methods 2005, which in a similar way presented software for iEMG decomposition, but (in my opinion) had a better integration of experimental data in the manuscript.

Validity of the findings

No comments

Additional comments

Overall, the manuscript is well written and provides a clear overview of the software. Apart from the suggestion to include more experimental data, the authors should consider a few additional comments:

- The cited review paper by Farina & Negro 2015 (and other work by that group) argues against the use of correlation between single pairs of motor units and suggests population-based analyses. I think this point should be emphasized, and possibly discussed as a future add-on to the software.

- It is a reasonable point that synchronization analysis usually requires expensive software, but the authors seem to overlook the fact that the same expensive software is typically required to obtain single motor unit spike trains (decomposition). In that sense, decomposition free-ware needs also to be developed in order to fulfill the ambition of enabling motor unit analysis for those without access to expensive software. This point should be discussed.

- In many articles using the CIS index, an additional parameter describing whether the peak in the histogram can be considered significant or not ("significant peak") is included. Here, significance is based on the peak in the Cusum. The authors may consider including this parameter as well.

- A prerequisite for valid estimation of synchronization is a critical number of decomposed spikes in the two spike trains. With too few spikes, a reasonable cross-histogram cannot be obtained. To the best of my knowledge no specific limit has been established, but I would suggest that the authors include typical number of spikes per spike train (used in previous studies) in the manuscript, in order to emphasize the importance to new users.

·

Basic reporting

The paper is well written but it is not a research paper. There are no hypotheses or results.

Experimental design

This is a technical paper that describes a software package that is likely to make a worthwhile contribution to investigations of motor unit firing behavior. However, it is not a research paper and so does not really seem to fall into the Aims and Scope of the journal. I'm afraid I don't have any previous experience with this journal to know if this lack is disqualifying. If it is, then perhaps the authors could consider the Journal of Neuroscience Methods or the IEEE Transactions on Neural Systems and Rehabilitation.

There is no original research, no research question, no investigation, and no methods.

Validity of the findings

There are no findings.

Additional comments

The paper is good and worthwhile, but it is not a research paper and therefore does not seem to fall within the scope of this journal.

---

## Round 0.2 · Minor Revisions

The reviewers have raised additional comments that need to be addressed before the manuscript can be accepted for publication.

·

Basic reporting

no comment

Experimental design

I would like to thank the authors for this strongly improved version. Most of my previous concerns have been addressed and I only have a few final comments.

126-127 Can you provide a reference for or very brief explanation of this "template matching technique"?

150-153 "if motor unit action potential trains are independent, then there is no relationship between the firing of one motor unit and the firing of the other and each latency or recurrence interval (e.g. -2 ms, +1 ms, or +6 ms) has an equal probability of occurring"
This statement in its current generality is not true, as I already indicated in my first review. For illustration just assume that both MUs fire independently at firing rates 20Hz and 30Hz, then it is very unlikely to observe a recurrence interval greater than, say, 100ms. In general, the distribution of the recurrence interval of two independently firing motor units will depend on the firing rates of the two motor units and not be uniform. I am unsure how this can be reconciled with the methods described here - maybe the distribution is approximately uniform for recurrence intervals below some threshold?

171-174 Thanks a lot for the improved description of this method. Just to make sure that I understand things correctly now: I assume "cumsum histogram" should be "cumsum graph" and refers to Fig. 2A? Then if I look for the 10%-90% range of the y axis in Fig. 2A I still end up with a really wide "peak", going from something around -0.08s up to maybe 0.02s. Is it just that the method does not work for this example (which would be helpful to mention) or do I still misunderstand something?

186-187 The same comment as above applies regarding the distribution of the recurrence interval of independently firing motor units.

185-198 What is the "shuffling" required for? Wouldn't it be equivalent to just compute mean and standard deviation of the original histogram data directly, without reshuffling? Also, the z-score is commonly defined as the difference of a sample from the population mean, divided by the standard deviation, and it is not identical to a confidence interval. In other words, a sample has z-score=2 if it is 2 standard deviations away from the mean. Under the assumption of a Gaussian distribution (is this justified here, i.e., is the amplitude distribution of bin counts Gaussian?), 95% of the data have z-score<=2 (asymptotically). The authors might want to revise this section to improve statistical rigor.

Validity of the findings

no comment

Additional comments

no comment

·

Basic reporting

-

Experimental design

My comments to the previous version of the manuscript have been addressed.

Validity of the findings

-

Additional comments

My comments to the previous version of the manuscript have been addressed.

·

Basic reporting

The paper is well written and self contained. Raw data is shared in an associated web page.

Experimental design

The paper is mostly a technical paper to present a software package. But it does present data used to test the software, The methods are described in sufficient detail.

Validity of the findings

The data provided are sufficient for the purpose of demonstrating the software, and they also serve to make the point that synchronization indices can vary considerably when computed by different methods.

Additional comments

26 "and investigate the differences in motor unit synchronization produced by different methods in a case study example." A single case study does not really constitute an investigation. Perhaps you could change this to something like "and present a case study showing how much the same synchronization index can differ when different methods are used to compute it."

348. "According to results from Keen et al. (2012), this 1 impulse per second difference represents 471% of the standard deviation of CIS in the FDS." For some reason, your results fall way, way outside the results reported by Keen et al. Therefore it probably doesn't make much sense to characterize the difference in terms of those results. Can you just use the 45% difference figure you used in the Abstract?

Read-me file. One possible point of confusion for readers of the motoRneuron.docx read-me file is that the cross-correlation histogram shown there (which is the one computed from the sample data in the code repository) is quite different from the one shown in Fig. 2 of the paper. First, the histogram in Fig. 2 has a relatively flat baseline count across its entire width, whereas the one in the read-me file falls off toward both ends. Presumably this was because the read-me histogram is based only on first-order intervals, whereas the one in Fig. 2 is based on first and higher order intervals. This should be explained in the read-me file.

Second, the central peak in Fig. 2 is compact and distinct as one often sees in short-term synchronization papers, whereas the central peak in the read-me histogram is diffuse and difficult to delineate visually. Perhaps this is the result of having too few recurrence intervals in the dataset, so that the statistical fluctuations in the bin counts are not sufficiently averaged out? In any event, some discussion about the difference between the central peaks in the two histograms is needed in the read-me file.

---

## Round 0.3 · accepted · Accept

The authors have adequately addressed the outstanding comments.